# Glioblastoma under Siege: An Overview of Current Therapeutic Strategies

**DOI:** 10.3390/brainsci8010015

**Published:** 2018-01-16

**Authors:** Mayra Paolillo, Cinzia Boselli, Sergio Schinelli

**Affiliations:** Department of Drug Sciences, University of Pavia, Viale Taramelli 12, 27100 Pavia, Italy; cinzia.boselli@unipv.it (C.B.); sergio.schinelli@unipv.it (S.S.)

**Keywords:** monoclonal antibodies, alkylating agents, nanoparticles, immunotherapy

## Abstract

Glioblastoma is known to be one of the most lethal and untreatable human tumors. Surgery and radiotherapy in combination with classical alkylating agents such as temozolomide offer little hope to escape a poor prognosis. For these reasons, enormous efforts are currently devoted to refine in vivo and in vitro models with the specific goal of finding new molecular aberrant pathways, suitable to be targeted by a variety of therapeutic approaches, including novel pharmaceutical formulations and immunotherapy strategies. In this review, we will first discuss current molecular classification based on genomic and transcriptomic criteria. Also, the state of the art in current clinical practice for glioblastoma therapy in the light of the recent molecular classification, together with ongoing phases II and III clinical trials, will be described. Finally, new pharmaceutical formulations such as nanoparticles and viral vectors, together with new strategies entailing the use of monoclonal antibodies, vaccines and immunotherapy agents, such as checkpoint inhibitors, will also be discussed.

## 1. Glioblastoma Classification

Gliomas are the most common primary central nervous system (CNS) tumors and, among these, half of all new diagnosis is represented by glioblastoma (GBM), the most malignant type of brain cancer with a poor prognosis and a median patient survival of approximately 18 months [1]. The histological classification and tumor grading is a critical step for GBM diagnosis and prognosis and the classification according to histological criteria by microscopic observation of specimen has been the main tool for pathologist and clinicians during the last decades.

Recently, the new World Health Organization (WHO) Classification of Tumors of the CNS [2] included both histological and molecular criteria to better integrate information from research and clinic, in order to achieve more accurate diagnosis. Several excellent papers have outlined the relevance and the impact of the most recent glioma classification on diagnosis, prognosis and therapy of this severe brain tumor [3,4]. Molecular subtyping, indeed, appears to be essential to identify subsets of patients that may be uniquely responsive to specific adjuvant therapies [5] and future therapies will be likely designed to target these molecular features.

In addition, as a result of genomic profiling and the Cancer Genome Atlas Project [6], more than 600 genes were sequenced from more than 200 human tumor samples, which revealed the extremely complex genetic profile of GBM and established a set of three main signaling pathways that are commonly altered: the p53 pathway, the receptor tyrosine kinase/Ras/phosphoinositide 3-kinase signaling pathway and the retinoblastoma (Rb) pathway. The consequences of alterations in these pathways are represented by uncontrolled cell proliferation, enhanced cell survival and infiltration skills, contemporarily conferring to the tumor cell the ability to escape from cell-cycle checkpoints and apoptosis [7].

### 1.1. Molecular Classification of Glioblastoma

Molecular alterations of gene expression patterns have been assessed between primary and secondary gliomas. Genetic alterations typical for primary GBM are epidermal growth factor receptor (EGFR) gene mutation and amplification, phosphate and tensin homologue (PTEN) mutations and chromosome 10q loss. In secondary GBM, isocitrate dehydrogenase 1 (IDH1) mutations, p53 mutations, and chromosome 19q loss are frequently found [4]. In addition, on the basis of mRNA expression analyses, four GBM subtypes have been identified (classical, pro-neural, neural, and mesenchymal), each with distinctly different patterns of disease progression, survival outcomes and response to therapy [6,8]. A subtype of GBM, accounting for nearly 10% of all glioblastomas, has been found to display isocitrate dehydrogenase 1 or 2 (IDH) mutations; mutant IDH proteins acquire the enzymatic ability to convert α-ketoglutarate (α-KG) to d-2-hydroxyglutarate (d-2-HG), eventually leading to aberrant DNA and histone methylation. This subtype mainly includes secondary glioblastomas, therefore sharing this mutation with lower grade astrocytomas, and is typically diagnosed in young adults. The prognosis for these patients, however, is normally more favorable than for IDH-wild-type glioblastoma, with a longer survival [2,9].

### 1.2. Epigenetic Analysis: DNA Methylation

DNA-methylation analysis is a useful tool to distinguish glioblastoma subgroups associated with specific epigenetic and genetic features. On this basis, four subgroups of adult glioblastoma have been identified, including an IDH mutant, and three IDH wild-type glioblastoma subgroups; the IDH mutant subtype has methylated O6-methylguanine methyltransferase (MGMT) promoter, displays a number of chromosomic aberrations and other alterations such as MYC activation, upregulation of the receptor tyrosine kinase (RTK)/Ras/PI3K pathway and mutations of genes encoding inhibitors of the G1/S cell-cycle checkpoint, including the Rb pathway [10,11].

The IDH wild-type glioblastoma subgroups have different DNA-methylation profiles and a fewer copy-number aberations. The so called “receptor tyrosine kinase I” (RTK I) glioblastomas are characterized by platelet-derived growth factor receptor A (PDGFRA) amplification and are mainly diagnosed in young subjects.

The “receptor tyrosine kinase II” (RTK II) and the “mesenchymal” subtypes are generally diagnosed in older patients (from 50 years of age) and the mesenchymal glioblastoma shows a mesenchymal gene-expression profile and is related to radioresistance and shorter survival [10].

Alterations of miRNA levels have been found in almost all cancer biology processes, including cell proliferation, migration, angiogenesis, and chemoresistance. Alterations of GBM miRNAs have been reported and a very recent study identified a panel of miRNAs that are likely to be dysregulated by genomic deletions and amplifications. In addition, the authors found that a miRNA acting as a tumor suppressor, miR-4484, is deleted in GBM, thereby leading to deregulation of a panel of genes involved in various cancer-related processes [12].

In the future, the need to distinguish between neoplastic cells and surrounding cells belonging to tumor niche, in the aim of finding molecular targets for glioma therapy, will represent a real challenge. Recently, two interesting papers have addressed this issue using single-cell analysis coupled to Next-Generation Sequencing (NGS) techniques. In the first study, single cell analysis of cells from the tumor core and surrounding tissue found a wide cellular variation in the tumor genome and transcriptome. Conversely, infiltrating GBM cells appeared to share a consistent gene signature between patients, suggesting the existence of common features underlying the infiltration mechanisms [13].

In another report [14] single cell-RNA-seq profile could distinguish between IDH-mutant astrocytoma and oligodendroglioma on the basis of their distinct tumor microenvironment (TME) and genetic signature. Enhanced proliferation of malignant cells, larger pools of undifferentiated glioma cells and an increase in macrophage over microglia expression programs in TME were related to tumor grade.

## 2. Therapy, State of the Art

Current standard therapy for GBM includes surgical resection, followed by radiation and co-administration of temozolomide (TMZ), an oral alkylating agent. Extensive surgical resection, though representing the most effective way to increase survival of GBM patients, is hardly feasible depending on tumor localization and infiltration, particularly when highly specialized brain areas are involved, such as those involved in the control of speech, motor function and senses. However, the highly infiltrative behavior of GBM makes surgery nearly ineffective, since tumor cells and glioblastoma stem cells (GSC) colonize the surrounding brain tissue causing relapses located even at distant brain sites [15]. Nevertheless, cutting-edge imaging techniques in both diagnostic and surgical phases make now possible a more aggressive surgical approach with limited side effects for patients [15]. The imaging techniques include functional magnetic resonance imaging (MRI) and diffusion tensor imaging (DTI), during preparatory phase, and ultrasound, computed tomography (CT) and MRI during surgery.

In addition to these techniques, the use of Gamma Knife radiosurgery (GKRS) has provided advantages especially for recurrent glioblastoma (rGBM). GKRS is a type of stereotactic radiosurgery capable of delivering a high dose of radiation to a tumor, sparing healthy surrounding tissue and is particularly used in rGBM. GKRS does not require open resection and allows minimization of radiation-induced necrosis to surrounding brain tissue [16]. The main disadvantage is represented by the fact that tumor areas not well detected by MRI cannot be reached by GKRS. In addition, radionecrosis and especially radiation-induced edema are strong drawbacks that have been reported in a high percentage of patients, nearly 30%, who received high radiation doses [17]. Interestingly, the concomitant administration of bevacizumab prolonged patient survival and significantly reduced detectable adverse radiation effects [18].

Hyperthermia is another strategy adopted in the treatment of GBM patients. Hyperthermia brings tumor tissue temperature to a range of 41–46 °C, thereby inducing physiological changes in tumor cells, such as protein misfolding, aggregation, alteration of signal transduction pathways, eventually leading to apoptosis [19]. The efficiency of hyperthermia treatment significantly depends on the temperature at the targeted tumor site, the period of exposure, and the features of cancer cells. Hyperthermia is generally achieved by the use of microwaves, infrared irradiation, ultrasound and tubes containing boiling water. Conversely, these systems suffer from limitations including unintended heating of healthy tissue with spread of heat by the blood, especially in highly vascularized tumors, and low diffusion of heat to the target sites [19]. Magnetic materials can be used to induce hyperthermia (magnetic hyperthermia, MHT) [20], but some undesirable side effects still remain. The use of nanoparticles (Nps) in combination with hyperthermia is a novel approach that allows controlled heating of targeted tumor tissue [20].

However, despite these technologies, differentiating between normal brain and tumor tissue continues to be a major challenge and, even with advances in surgical resection, the prognosis for GBM patients remains poor.

On the pharmacological therapy side, great efforts have been made to develop new pharmacological tools leading to a series of clinical trials recently completed or currently ongoing (Table 1 and Table 2).

The classical clinical practice includes TMZ treatment for GBM patients. TMZ is a derivative of the alkylating agent dacarbazine and is active against human cancers such as melanomas and astrocytomas by inducing cell cycle arrest at G2/M phase and eventually leading to apoptosis [21,22,23]. Particularly, TMZ acts by methylating adenine and guanine residues to form N3-methyladenine, N7-methylguanine and O6-methylguanine. The methylated bases can be repaired by DNA repair enzyme systems that reverse guanine methylation induced by TMZ, thus preventing apoptosis initiation. The mechanisms of resistance to TMZ depend on different DNA repair systems, among which methylguanine methyltransferase (MGMT) aroused great interest [24,25]. MGMT is an enzyme involved in DNA repair processes that counteract mutagenesis from alkylating agents [24,25]. High levels of MGMT activity in cancer cells lead to a phenotype that is resistant to alkylating agents and MGMT is likely to play an important role in therapeutic failure. Indeed, epigenetic silencing of the MGMT gene by promoter methylation is associated with loss of MGMT expression [26] and diminished DNA-repair activity. Promoter methylation has also been associated with longer overall survival in patients with glioblastoma treated with carmustine or TMZ [26].

To overcome this resistance mechanism, the combination of capecitabine and TMZ was designed. Capecitabine is a prodrug of the pyrimidine analog 5-FU (5-fluorouracil), which is enzymatically converted to 5-FU and leads to inhibition of MGMT repair activity, probably through depletion of MGMT protein and mRNA [27,28,29,30]. Clinical trials including the association capecitabine/TMZ are still ongoing (Table 2).

## 3. New Strategies

### 3.1. Nanoparticles

Nanoparticles (Nps) are structures in the nanometer size range having different structures and properties and, in general, are widely studied for biomedical applications [31]. In GBM treatment several nanoparticle types are under investigation with different application strategies.

Lipid carriers are bilayered vesicles composed of phospholipid membranes (liposomes). Liposome phospholipids are mainly represented by biocompatible fatty acids, such as phosphatidylcholine and ethanolamine, and display on their surface both hydrophilic and hydrophobic regions [31]. These vesicles are very flexible tools, can be loaded by a variety of drugs and are highly biocompatible. Paclitaxel is a classical and effective chemotherapeutic drug that is unable to cross the brain barrier (BBB) and for this reason cannot be used for brain tumor treatment. In the aim to make paclitaxel cross the BBB, liposomes have been loaded with the drug and used for GBM therapy [32]. Also other drugs such as peptides, monoclonal antibodies, siRNA and other molecules, which otherwise would not be able to pass the BBB, are under investigation for use in brain tumor treatment as liposome formulations.

Inorganic Nps can be utilized for diagnostic and therapeutic purposes. Magnetic Nps (MNps) are the most common Nps used in biomedical applications due to their high biocompatibility; injectable super paramagnetic iron oxide Nps (SPION) can be also used in MRI with lower toxicity effects and higher sensitivity compared to conventional contrast agents [33]. In addition, SPIONs display interesting features such as high intratumoral penetration and controlled heating: hyperthermia, in fact, besides destroying tumor cells, can also be used to deliver drugs to tumors such as GBM. Similarly, gold Nps can be used both as contrast agent for MRI and for photothermal therapy. Particularly, gold nanorods (GNRs) have been used for thermal ablation of GBM cells, in vivo [20].

An interesting application is the use of Nps in combination with siRNAs; it is known that stimulation of the thyrosin kinase receptor c-Met induces proliferation of glioma cells and resistance to chemotherapy; intravenous administration of polyethylene glicol (PEG) Nps loaded with c-Met siRNA reduced c-Met expression and powerfully inhibited cell proliferation and resistance to chemotherapeutic agents in GBM cells [34].

Glioblastoma stem cells (GSCs) are currently held responsible for GBM infiltration in the brain tissue, proliferation and resistance to chemotherapy [35]. For these reasons, they appear as a target of primary importance in the fight against GBM; in GBM patients the micro RNA miR-1 has been found to be deregulated and delivery of miR-1 by Nps has been found to efficiently target GSC thereby reducing cell migration and proliferation [36].

### 3.2. Targeted Therapies and Immunotherapy

It was once thought that the central nervous system was devoid of normal immunologic function and lymphatic vessels [37]. Indeed, some CNS features support this theory, such as the blood-brain barrier structure, which allows for selective entry of immune cells from the peripheral blood into the brain tissue, and the low number of circulating T cells in the CNS. Concerning the lack of lymphatic vessels within the CNS, in 2015 an important finding changed this perspective: meningeal lymphatic vessels were identified in mouse brain and these vessels were found to be a path for cerebrospinal fluid drainage, demonstrating for the first time the presence of a functional lymphatic system in the central nervous system [38].

Furthermore, under physiologic conditions, the brain hosts several populations of immune cells such as microglia, which stem from hematopoietic cells and colonize CNS during embryonic development. Microglial cells migrate to inflammatory sites in the CNS, activate and behave like phagocytes or antigen presenting cells, secreting cytokines and chemokines [37].

This picture is different in GBM patients, who display a slight but detectable immune suppression condition, compared to the general population, and low adaptive immune responses. In addition, the tumor microenvironment is rich in immunosuppressive factors secreted by the tumor, like transforming growth factor beta (TGF-β) and vascular endothelial growth factor (VEGF) that suppresses cytotoxic T cell activity [39].

In this background, different approaches are under investigation: in targeted therapies, immune cells or antibodies directed against tumor antigens are given to patients and this approach does not require activation of patients’ immune system. Over recent years several molecular targets have been identified and tested in GBM therapy, with controversial results (Table 1).

In GBM the so-called EGFRvIII mutation is frequently found; this mutation arises from a deletion of 267 amino acids of the extracellular domain leading to the expression of mutants with a unique extracellular domain [40]. This mutant EGFRvIII is ligand independent, constitutively active and, because of its important mitogenic effects, is related to a short survival.

Some monoclonal antibodies (mAbs) binding to EGFRvIII have been developed but, despite the relevant results of antibody-based therapy in the treatment of other cancer types such as renal carcinoma, breast cancer, melanoma, and hematologic cancers, these results have not been equally successful in GBM [41]. Among the antibodies developed to target wtEGFR and EGFRvIII, cetuximab, panitumumab, and nimotuzumab, which bind the extracellular EGFR domain, were included in clinical trials that were recently completed with varying results (Table 1). In a phase II study patients were stratified on the basis of EGFR gene amplification status and were administered cetuximab intravenously [42]. Cetuximab had little effect and the median overall survival was 5 months, showing no significant correlation between EGFR status and response or overall survival [36]. Other clinical trials involving similar antibody-based therapies have been equally unsuccessful, even when nimotuzumab was administered with concurrent radiotherapy [40]. 

Indeed, the first mAb tried in the therapy of GBM patients was bevacizumab. Bevacizumab is a humanized monoclonal antibody that binds to the different forms of the vascular endothelial growth factor (VEGF). Bevacizumab is currently used in the therapy of a variety of tumor types and is also in trial for GBM therapy. However, in recent, randomized, double-blind, placebo-controlled studies designed to evaluate first-line use of bevacizumab together with the standard therapy, chemoradiation and TMZ, no increase of median overall survival was reported. It should be also noted that patients were stratified by MGMT promoter methylation and other gene mutations but no group showed benefits from bevacizumab treatment [43] with, in contrast, a marked increase of adverse events typically related to bevacizumab, such as deep vein thrombosis, gastrointestinal perforations, and hemorrhage [43].

New insights could still come from ongoing clinical trials but at the moment classical antibody-based approaches are making time.

A promising possibility is represented by the use of mAbs linked to cytotoxic molecules that specifically target cancer cells, thereby delivering drugs or toxins; these mAbs are named antibody drug–conjugate (ADC) and are under investigation for the use in GBM patients [44]. Immunotoxins such as Pseudomonas aeruginosa exotoxin A (PE) and diphtheria toxin (DT) have been tested as ADC in GBM patients but the clinical trials did not produce the expected results. Other strategies include anti-EGFR mAbs conjugated with drugs not suitable to use in therapy due to their high toxicity, such as maytansine and monomethyl auristatin F (MMAF), both inhibitors of microtubules assembly. Clinical trials, including the use of these ADCs alone or in combination with temozolomide, are in phases I and II, and preliminary results indicate a good and selective uptake of these conjugates by tumor cells and a tolerable toxicity profile, limited to retinal toxicity [44].

Another interesting target is the αvβ3 integrin receptor, which is highly expressed in glioblastoma and can be targeted by mAbs or by integrin receptor ligands conjugated to cytotoxic drugs; preclinical studies are giving promising results [45].

Immunotherapy offers a different approach from chemotherapy, targeted therapy, radiation and surgery, raising new hopes, particularly for immune checkpoints inhibitors. T lymphocytes can recognize antigens expressed by cancer cells but, conversely, immune checkpoints, particularly programmed cell death (PD)-1 receptor and its ligand (PD-L1), can suppress the activity of T lymphocytes by inducing apoptosis in activated immune cells [46,47]. The expression of PD-L1 has been demonstrated in glioma cell lines and tumor tissues. Quite interestingly, it was reported that PD-L1 expression was significantly greater at the edges of the tumors than in the tumor cores, thereby leading to the formation of a sort of barrier between the tumor cells and cytotoxic T cells that has been defined as “molecular shield”. The strategy of blocking PD-1 and PD-L1 by mAbs has given encouraging results in the treatment of other cancer types; pembrolizumab and nivolumab, targeting PD-1, were approved by Food and Drug Administation (FDA) and European Medicines Agency (EMA) for advanced melanoma therapy in late 2014 and for non-small cell lung cancer (NSCLC) therapy in March 2015 [48,49].

The growing interest in immune checkpoint inhibitors and the encouraging results in the treatment of these cancer types have driven in recent years the attention of researchers towards the use of anti PD-1 and PD-L1 mAb in GBM therapy. Preclinical studies in mice have given promising results and a number of clinical studies involving pembrolizumab and nivolumab alone or in combination with other agents (bevacizumab, temozolomide) are starting or still ongoing and results will be available in a few years.

Also the very new therapies involving the use of genetically modified T cells expressing chimeric antigen receptors (CARs) may represent a new offensive strategy in the GBM siege. These T cells are modified to specifically recognize GBM antigens such as IL13Rα2, HER2, EphA2, and EGFRvIII, the first antigens to be tried in this type of immunotherapy [50,51], and phase I trials in humans are ongoing. In one reported case, autologous CAR T cells targeting IL13Rα2 induced a complete response in a patient with recurrent multifocal glioblastoma, with dramatic improvements in quality of life and a return to normal life activities, allowing 7,5 disease free months survival [52].

### 3.3. Oncolytic Viruses

Oncolytic viruses (OV) specifically target cancer cells; OV have been designed to take advantage of tumor-specific mutations, or signaling pathways that are constitutively activated in tumors, and are selected to enter tumor cells overexpressing tumor antigens [53]. The infected tumor cells thereby undergo apoptotic processes or necrosis and eventually cell death.

Several OV are under investigation for high-grade glioma therapy and, among these, two types of modified Herpes simplex virus 1 (HSV1) and adenovirus (AdV) are in clinical trial. Indeed, a variety of oncolytic HSV mutants such as R3616, HSV-1716, hrR3, G207, and G47Δ have been designed for targeting glioblastoma. All these mutants display deleted or mutated viral genes, thus reducing neurotoxicity while not affecting infection of dividing cells, particularly those with activated Ras pathway signaling [54]. In G207 mutant the ICP6 gene encoding the viral ribonucleotide reductase is inactivated in order to increase safety, since this enzymatic function is recovered in proliferating glioma cells but not in non-tumor quiescent cells [55]. In addition, G207 has been found to interfere with replication of p16 tumor suppressor gene defective cells [56], such as glioma cells, and is now in phase II trial for rGBM therapy.

AdV is a non-enveloped virus with a double-stranded, linear DNA genome. There are multiple engineered versions under investigation, including DNX-2401 and ADV-TK that are in clinical trial. DNX-2401 mutation allows for replication only in cells bearing Rb tumor suppressor deletion [55]. The tumor specificity is given by the presence of the cyclic arginine/glycine/aspartic acid (RGD) peptide into the viral capsid region responsible for attachment to host cells; the RGD peptide directs the virus towards RGD-binding integrins, which are highly expressed in gliomas. Preliminary recently published results from trials show that therapy with DNX-2401 in combination with temozolomide or interferon-γ is well tolerated and shows significant therapeutic activity [57,58].

ADV-TK is an adenoviral vector engineered to express the Herpes thymidine kinase gene followed by administration of an anti-herpetic prodrug (ganciclovir—GCV). A randomized phase II clinical trial demonstrated progression-free survival and overall survival benefit associated with ADV-TK gene therapy, which was also shown to be safe [59].

Another interesting therapeutic strategy in clinical trial is represented by the use of Toca 511 and Toca FC [60]. These agents act in combination and, considering the rationale of this therapy, are expected to display very limited toxic effects. Toca 511 (vocimagene amiretrorepvec) is an injectable retroviral replicating vector encoding an enzyme, cytosine deaminase (CD), derived from yeast and not expressed by humans. CD is a prodrug activator that is selectively delivered to cancer cells that, in turn, acquire the ability to synthesize CD protein. This enzyme acts on a substrate, 5-fluorocytosine (5-FC), a prodrug that is inactive in human but is converted in 5-FU, a widely used anti-cancer agent, by the CD protein. Toca FC is an oral formulation of 5-fluorocytosine (5-FC) that is easily absorbed, is able to cross the blood-brain barrier and to diffuse into the cancer cells [60]. Toca 511 and Toca FC are currently in trial in patients with recurrent high-grade glioma (Table 2) and are under investigation in a pivotal Phase 3 trial.

### 3.4. Vaccines

The EGFRvIII mutation occurs in roughly 20–30% of all glioblastomas and its potential immunogenicity has raised substantial interest in the neuro-oncology field; the studies aimed at finding new targets for glioblastoma therapy resulted in the development of rindopepimut, a peptide vaccine containing the amino acid sequence found in the EGFRvIII deletion mutation conjugated to keyhole limpet hemocyanin. In four separate glioblastoma trials, ACTIVATE, ACT II, ACT III, [61] rindopepimut given in combination with temozolomide was well tolerated and a consistent progression-free survival, in the range of 15 months from diagnosis, and overall survival of 24 months from diagnosis was found, which compared favorably with patient cohorts who received standard treatment. In the Phase II ReACT trial for recurrent glioblastoma, rindopepimut was given in combination with bevacizumab and a small number of patients with bulk disease appeared to benefit from vaccine therapy, when combined with bevacizumab [62,63]. Afterwards, the ACT IV was designed as a randomized, placebo-controlled, phase III clinical trial to assess whether the co-administration of rindopepimut and standard temozolomide therapy increased overall survival compared with temozolomide alone in patients with newly diagnosed EGFRvIII-expressing glioblastoma. In spite of previous promising results, final data analysis of this study demonstrated no survival benefit for patients with EGFRvIII-positive glioblastoma who received rindopepimut with temozolomide versus those who received a control, with median overall survival of 20.4 months compared with 21.1 months for the control arm [64].

Other glioblastoma vaccines are under investigation in single-arm and randomized Phase II trials and results will be available in a short time.

## 4. Conclusions

Evidence coming from preclinical studies and clinical trials indicate that strategies to counteract GBM aggressiveness are multiplying and that knowledge of GBM mutations and genetic profiling can highlight possible molecular targets for therapies. It is also clear that traditional therapies such as surgery, chemotherapy and radiation still remain the first line approaches to GBM and hopes to improve life quality and overall life expectancy of GBM patients mainly lie, at the moment, in the new immunotherapy strategies. In the coming years a number of clinical trials will give fundamental insights about these therapeutic approaches and, contemporarily, basic research must pursue the objective of discovering new mechanisms and new targets to help and direct clinical research.

## Figures and Tables

**Table 1 brainsci-08-00015-t001:** Major studies completed during the last three years (2015–2017).

Drug(s)	Phase	Conditions	Completion Date	National Clinical Trials (NCT) Number
^1^ Bevacizumab ^2^ Amgen 386	1,2	Glioblastoma Multiform	January 2017	NCT01290263
^1^ Bevacizumab ^3^ LBH589	1,2	Malignant Glioma	December 2015	NCT00859222
^1^ Bevacizumab ^4^ Onartuzumab	2	Glioblastoma	January 2016	NCT01632228
^1^ Bevacizumab ^5^ Carmustine	2	Glioma	December 2015	NCT00795665
^5^ Temozolomide ^6^ ABT-888	1,2	Recurrent Glioblastoma	December 2016	NCT01026493
^6^ BSI-201 ^5^ Temozolomide	1,2	Glioblastoma	June 2015	NCT00687765
^5^ Temozolomide ^7^ BIBW 2992	2	Glioma	May 2016	NCT00727506
^7^ Cetuximab	1,2	Malignant Glioma	November 2016	NCT02855086
^8^ Tivozanib	2	Glioblastoma	May 2016	NCT01846871
^8^ Cediranib ^5^ Lomustine	3	Recurrent Glioblastoma	September 2016	NCT00777153
^9^ PLX3397	2	Recurrent Glioblastoma	January 2015	NCT01349036
^10^ CAR-T cell (chimeric T cell receptors artificial T cell) Immunotherapy	1,2	GD2 Positive Glioma CAR-T Cell Immunotherapy	August 2017	NCT03252171
^10^ ICT-107	2	Glioblastoma Multiforme	December 2015	NCT01280552
^10^ PEP-3 vaccine ^11^ Sargramostim ^5^ Temozolomide	2	Glioblastoma Multiforme	November 2016	NCT00643097
^10^ CDX-110 ^11^ GM-CSF ^5^ Temozolomide	2	Malignant Glioma	May 2016	NCT00458601
^10^ Rindopepimut (CDX-110) ^11^ Sargramostin (GM-CSF)	3	Glioblastoma	November 2016	NCT01480479

^1^ antiVGEF (vascular endothelial growth factor)-antibody; ^2^ angiopoietin/Tie pathway inhibitor; ^3^ pan-DAC (histone deacetylase) inhibitor; ^4^ HGF (hepatocyte growth factor)-inhibitor; ^5^ alchylating agent; ^6^ PARP (poly ADP ribose polymerase) -inhibitor; ^7^ EGFR (epidermal growth factor receptor)-inhibitor; ^8^ VEGF-TKI (tyrosine kinase) inhibitor; ^9^ CSF1R (colony stimulating factor 1 receptor) -inhibitor; ^10^ vaccine; ^11^ hematopoietic growth factor.

**Table 2 brainsci-08-00015-t002:** Major ongoing clinical trials based on pharmacological treatment(s) of malignant glioma.

Drug(s)	Phase	Conditions	Estimated Completion Date	NCT Number
^1^ Bevacizumab	1,2	Glioblastoma Multiform	January 2018	NCT01811498
^1^ Bevacizumab	2	Glioblastoma	May 2018	NCT02157103
^1^ Bevacizumab ^2^ Temozolomide Dietary Supplement Vitamin C	1,2	Malignant Glioma	March 2020	NCT01891747
^1^ Bevacizumab ^2^ Temozolomide	2	Glioblastoma	January 2019	NCT01149850
^1^ Bevacizumab ^2^ Temozolomide	3	Recurrent Glioblastoma	July 2023	NCT02761070
^3^ Cetuximab	1,2	Glioblastoma	June 2019	NCT02861898
Cerebraca Wafer ^2^ *n*-butylidenephthalide (BP)	1,2	Recurrent High Grade Glioma	August 2019	NCT03234595
^2^ TH-302 ^1^ Bevacizumab	2	Glioblastoma	July 2018	NCT02342379
^2^ Temozolomide Metformin	2	Glioblastoma	December 2018	NCT03243851
^2^ VAL-083 (Dianhydrogalactitol) ^2^ Temozolomide or ^2^ Lomustine or ^2^ Carboplatin	3	Glioblastoma Multiform	August 2019	NCT03149575
^4^ Capecitabine ^2^ Temozolomide	1,2	Glioblastoma Multiform Glioblastoma	June 2021	NCT03213002
^6^ Cisplatin ^2^ Temozolomide	2	High-grade Gliomas	December 2017	NCT02263105
^7^ SGT-53 ^2^ Temozolomide	2	Recurrent Glioblastoma	December 2019	NCT02340156
^8^ Cediranib Maleate ^9^ Olaparib ^1^ Bevacizumab	2	Recurrent Glioblastoma	October 2019	NCT02974621
^10^ Neratinib ^3^ CC-115 ^11^ Abemaciclib ^2^ Temozolomide	2	Glioblastoma	May 2021	NCT02977780
^12^ Nivolumab ^1^ Bevacizumab ^13^ Ipilimumab	3	Recurrent Glioblastoma	January 2018	NCT02017717
^14^ Toca 511 ^14^ Toca FC ^2^ Lomustine ^2^ Temozolomide ^1^ Bevacizumab	2,3	Glioblastoma Multiform	September 2019	NCT02414165
^14^ VB-111 ^1^ Bevacizumab	3	Glioblastoma	December 2017 (primary outcome)	NCT02511405
Dendritic cell vaccine plus ^2^ Temozolomide	1,2	Glioblastoma Multiform	December 2019	NCT02649582
alpha-IFN ^2^ Temozolomide	3	Glioblastoma	December 2017	NCT01765088
CIK (Cytokine-Induced Killer Cells) ^2^ Temozolomide	4	Advanced Malignant Gliomas	July 2030	NCT02496988
^15^ DNX-2401 With Interferon Gamma (IFN-γ)	1	Recurrent Glioblastoma or Gliosarcoma Brain Tumors	August 2018	NCT02197169
^15^ DNX-2401 ^16^ Pembrolizumab	2	Brain Cancer Glioma Glioblastoma	June 2020	NCT02798406

^1^ VGEF-inhibitor; ^2^ alchilating agent; ^3^ EGFR-inhibitor; ^4^ Thymidylate synthase inhibitor; ^5^ mTOR (mammalian target of rapamycin)-inhibitor; ^6^ DNA-binding inhibitor; ^7^ liposome encapsulating the wtp53 DNA sequence; ^8^ VEGFR inhibitor; ^9^ PARP inhibitor; ^10^ tyrosine kinase inhibitor; ^11^ Dual Inhibitor of CDK4 (cycline dependent kinase) and CDK6; ^12^ anti PD-1R (programmed death receptor) antibody; ^13^ anti CTLA4 (cytotoxic T-lymphocyte antigen 4)-antibody; ^14^ oncolytic virotherapy; ^15^ oncolytic adenovirus; ^16^ anti PD-1.

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
