# Peer review of "Glioblastoma under Siege: An Overview of Current Therapeutic Strategies"

_brainsci, 2018, doi:10.3390/brainsci8010015_

Round 1

Reviewer 1 Report

This is a nicely written review of GBM

Reviewer 2 Report

Glioblastoma (GBM) is one of the most lethal and aggressive brain cancer types, and none novel therapeutic strategies have been developed successfully for years. In this manuscript, the authors summarize the classification standards for GBM diagnosis, especially the emerging key markers which have been incorporated into the definitions of diagnostic entities, such as IDHs; introduce the current therapies used on patients’ treatment; and highlight the new and potential strategies including nanoparticle utilization and immunotherapy. Overall, this is a nice review, which would help researchers easily understand the current progress of basic and clinical research on GBM.

There are several arguable descriptions and a few text issues.

1.   P. 2, line 50: “EGFR gene mutation and amplification” would be more appropriate than “EGFR overexpression”.

2.   P. 2, line 68: the full name of MGMT “O6-methylguanine methyltransferase” is needed. Therefore, “O6-methylguanine methyltransferase” in line 144 is deletable.

3.   P. 3, line 140: “and is active against” would be better than “is active against”. And by stating TMZ is active against melanomas and astrocytomas, more references need to be cited, which can be found through reference 21.

4.   P. 4, line 146: “negative” and “it is” allow deletion.

5.   P. 7, line 183: “chemoherapeutic” should be “chemotherapeutic”.

6.   P. 7, line 199: “loaded by” is supposed to be “loaded with”.

7.   P. 8, line 232 and 233: the first sentence is removable. The correlation between GBM and EGFR has been introduced in section 1.1.

8.   P. 8, line 234: “leads to” should be replaced with “arises from”.

9.   P. 9, line 280 and 297: at least one of “Quite interestingly” is changeable.

Author Response

We thank the reviewer for his careful attention to our manuscript. We have revised and corrected it completely according to his suggestions.

Reviewer 3 Report

This is a review article in which the authors intended to overview recent advances in emerging treatments for glioblastoma. There are several problems that I ask the authors to address.

1.    Line143. Different methyl adducts of DNA are repaired by different pathways. N3mA and N7mG are primarily repaired by base excision repair pathway.

2.    Line149-150: Please be sure that the statement that 5-FU induces MGMT promoter methylation is correct. The reviewer cannot find evidence for this, and wonder if 5-FU reduces/depletes MGMT protein levels.

3.    Lines 153 and 163. The reviewer wonder if “Table 1” is an error and this should be Table 2.

4.    The Immunotherapy section uses a large space for describing approaches that use therapeutic antibodies. However, antibody drugs such as cetuximab and bevacizumab are usually not considered as immunotherapy. These are rather molecularly targeted therapies and as for bevacizumab, this is considered as an anti-angiogenic therapy as well. Subheading of “Immunotherapy” thus does not seem appropriate.

5.    Generally eliciting anti-tumor immune response, oncolytic virotherapy (OV) is an emerging modality that is drawing much attention, so probably is worth mentioning. (Toca511/Toca FC can be discussed together as these both use replication competent viruses). But if the authors want to focus on pharmacological (non-biological) therapies, exclusion of OV from the manuscript may be mentioned.

6.    Vaccination with tumor specific peptides such as a recently reported ACT IV study may be discussed in the manuscript.

7.    Line243. Anti-EGFR antibodies are not found in Table 1.

8.    Line243. Patients “with recurrent malignant glioma”.

9.    Line303-304. I do not understand what “GBM is positively under siege” means.

10. Both Tables are not organized / thought out well and the current forms are not very informative. Cannot Table 2 use subdivision based on the types of therapy? Is showing doses necessary in Table 1? Cannot footnotes be used to help readers know molecular targets or some more unfamiliar drugs better? Form of presentation also should be improved.

11. There are a number of typos, such as immunotherapic (line21), cromosomic (line 68) and thyrosine (line 198) among others.

Author Response

Reply to the reviewers

Reviewer 1

We agree with the reviewer that some issues should have been better clarified and the manuscript has been corrected accordingly to his suggestions.

1. The other DNA repair mechanisms have been discussed and references added.

2. The statement is correct, MGMT protein levels vary following 5FU treatment and references have been added to better clarify the issue (ref. 27-30).

3. The error has been corrected.

4. The title of the paragraph has been corrected and a new "immunotherapy" heading has been added.

5. Oncolitic viruses have been briefly discussed and a paragraph has been added.

6. The ACT IV study has been cyted in the "vaccines" paragraph.

7, 10. The tables have been modified to be, hopefully, clearer.

8, 9. The phrases have been modified.

11. Typos have been corrected.

Round 2

Reviewer 3 Report

Regarding my previous comment #1, the authors state that "MGMT induces DNA mismatch repair system (MMR) and base excision repair (BER) pathways". I do not think this is correct.

The authors did not understand and address my previous comment #2. I asked if the statement that "5-FU induces MGMT promoter methylation" is correct.

Author Response

reply to Reviewer 1:

- We have modified the statement, we agree that the verb "induces" could be misleading.

- We have investigated the issue, according to the reviewer hint, and we have found that well founded data at molecular level only concern MGMT protein and mRNA expression. The sentence has been changed accordingly.